# The Role of Luminal Apposing Metal Stents on the Treatment of Malignant and Benign Gastric Outlet Obstruction

**DOI:** 10.3390/diagnostics13213308

**Published:** 2023-10-25

**Authors:** Mihai Rimbaș, Kar Wai Lau, Giulia Tripodi, Gianenrico Rizzatti, Alberto Larghi

**Affiliations:** 1Gastroenterology Department, Colentina Clinical Hospital, Carol Davila University of Medicine, 020125 Bucharest, Romania; mihai.rimbas@umfcd.ro; 2Department of Gastroenterology, Royal Stoke University Hospital, Stoke-on-Trent ST4 6QG, UK; kar.lau@uhnm.nhs.uk; 3Digestive Endoscopy Unit, Fondazione Policlinico Universitario A. Gemelli, IRCCS, 00168 Rome, Italy; giulia.tripodi91@gmail.com (G.T.); gianenrico.rizzatti@gmail.com (G.R.)

**Keywords:** gastric outlet obstruction, surgical gastrojejunostomy, duodenal stenting, lumen-apposing metal stents, LAMS

## Abstract

Gastric outlet obstruction (GOO) is a clinical syndrome traditionally managed by surgical gastrojejunostomy or enteral stenting. The surgical approach is burdened with a high rate of adverse events (AEs), while enteral stenting has a limited long-term clinical effectiveness, with the need for repeat procedures. The availability of lumen-apposing metal stents (LAMSs) has resulted a shift in the treatment paradigm of GOO. Indeed, endoscopists are now able to create a stable anastomosis between the stomach and small bowel under endosonographic guidance. EUS-guided gastro-enteroanastomosis (EUS-GE) has the theoretical advantage of a durable luminal patency resulting from stent placement away from the site of obstruction, free from surgical-related AEs. This approach could be especially valuable in terminally ill patients with a limited life expectancy. The present paper reviews procedural techniques and clinical outcomes of EUS-GE in the context of both malignant and benign GOOs.

## 1. Introduction

Gastric outlet obstruction (GOO) is a clinical syndrome caused by either a benign or malignant mechanical impediment to gastric emptying. This mechanical obstruction can be caused by both benign and malignant conditions, caused either by luminal pathology or extrinsic compression of the distal stomach, pyloric channel, or duodenum. Malignant causes of gastric outlet obstruction commonly include advanced-stage gastric cancer, pancreatic cancer, duodenal cancer, and metastatic tumors to the stomach or duodenum. Benign causes may include peptic ulcer disease, gastric volvulus, pyloric stenosis, Crohn’s disease, and post-surgical complications.

The obstruction prevents the normal passage of fluids, food, and gastric secretions from the stomach into the small bowel. Subsequent symptoms include nausea, vomiting, early satiety, abdominal pain, and weight loss, which can have profound consequences on a patient’s quality of life and overall health [1]. Patients may experience significant weight loss, malnutrition, dehydration, electrolyte imbalances, and deterioration of overall health. In addition, persistent vomiting and inadequate oral intake can lead to aspiration pneumonia and other respiratory complications [2].

Thus, symptomatic GOO represents an indication for treatment regardless of the size of stenosis and type of disease—benign or malignant [3]. GOO has traditionally been managed by surgical gastrojejunostomy or enteral stenting. The surgical approach, which can also be performed laparoscopically, is able to restore the patency of the gastrointestinal tract for the long term; however, it is burdened with a high rate of adverse events (AEs). Although enteral stenting is efficacious for the majority of patients, with a low incidence of procedural AEs, long-term self-expandable metal stents (SEMS) placement (>3 months) leads to a significant reintervention rate due to tumor ingrowth or overgrowth. Other treatment options that did not gain widespread use were the NOTES approach [4,5] and the use of magnets [6,7,8] in order to create a bypass anastomosis between the gastric lumen and the small bowel distal to the obstruction.

In the last decade, the treatment paradigm for GOO has evolved with the availability of lumen-apposing self-expanding metal stents (LAMS). LAMSs simulate surgery by permitting endoscopists to create a stable anastomosis between the stomach and small bowel under endosonographic guidance [9]. EUS-guided gastro-enteroanastomosis (EUS-GE) carries the theoretical advantages of durable luminal patency by stent placement away from the site of obstruction, avoiding surgical-related AEs, especially in terminally ill patients with a limited life expectancy. The EUS-GE approach is, however, contraindicated in the presence of significant ascites that cannot be controlled pre-procedurally, in the case of malignant diffuse infiltration of the gastric or jejunal walls, or in the case of extensive peritoneal carcinomatosis.

In the present paper, we will review procedural techniques, post-procedure management, and clinical outcomes of EUS-GE in the context of both malignant and benign GOO.

## 2. Luminal Apposing Metal Stents

Lumen-apposing metal stents represent the first accessories specifically designed to be utilized under endosonographic guidance. They are self-expanding fully covered metal stents made of nitinol with a shape-memory alloy, which permits the stent to be compressed for endoscopic delivery and then expand to its predetermined configuration once deployed. The stent is designed to anchor itself within the opposing luminal walls, providing stability and preventing migration. This is achieved through a dumbbell design consisting of an inner lumen that serves as a conduit for drainage of luminal contents and two large-diameter flanges facilitating secure apposition between the luminal walls.

In 2012, Binmoeller and Shah [9] were the first to report the successful creation of a gastroenterostomy in a pig model by delivering, under EUS guidance, a lumen-apposing metal stent (15 mm in diameter and 10 mm in length, the AXIOS™ stent, Xlumena Inc., Mountain View, CA, USA) attaching the gastric and enteric walls. Furthermore, electrocautery-enhanced LAMSs (EC-LAMSs) have been mounted on a delivery device with an electrocautery tip with cystotome capability. This design allows for direct transmural access without prior needle puncture, and tract dilation using a pure cut current, rendering fistula creation and stent placement as a one-step procedure [10].

There are three EC-LAMSs available in the market: The Hot-Axios™ stent, the Niti-S™ Hot-Spaxus™, and the HANAROSTENT Hot Plumber™ Z-EUS IT™. The Hot Axios was the first developed EC-LAMS by Xlumena Inc. (Mountain View, CA, USA), which was subsequently purchased by Boston Scientific (Marlborough, MA, USA) in 2015 [11]. The stent delivery catheters can be of 9 Fr or 10.5 Fr and accommodate stents with inner diameters of 6 to 20 mm and saddle lengths between flanges of 8 to 15 mm (Figure 1A). The handle of the AXIOS delivery system is designed to be Luer-locked onto the echoendoscope inlet port of the instrumentation channel, and the stent release is fully controlled by the endoscopist. The advancement and deployment of the stent device is performed in a series of four dedicated steps, mainly under sonographic control, with fluoroscopic and endoscopic guidance, at the discretion of the operator. The release mechanism faces a hard stop between distal and proximal stent flange deployment. On the delivery catheter, located at the level of both ends of the preloaded stent, there are radiopaque markers for fluoroscopic control, while an endoscopically visible black marker is added to confirm the possibility of proximal flange release after correct stent apposition. All these features were added to prevent stent misdeployment.

The Niti-S™ Hot-Spaxus™ (Taewoong Medical, Busan, Republic of Korea) is the second developed EC-LAMS, which has some differences from the Hot-Axios™. Flares provide accommodative apposition regardless of the wall thickness ensuring adaptable lengths ranging from 7 to 20 mm. The stent is available in three body diameters (8 mm, 10 mm, and 16 mm, with flange diameters of 23, 25, and 31 mm, respectively) and is delivered through a 10 Fr catheter that has a channel in which a 0.035″ guidewire can be preloaded (Figure 1B). By using a pure cutting current of 80–120 Watts, the electrocautery device enters into the target structure without the need for prior tract dilation. Except the electrocautery tip, the delivery system is otherwise similar to that of other standard self-expanding metal stents, and its deployment requires the help of an assistant. In a small study of 58 patients, the use of this EC-LAMS resulted in high technical and clinical success rates for various interventional EUS indications, with adverse events similar to that of a control group of patients treated with the Hot Axios device [14].

The HANAROSTENT Hot Plumber™ Z-EUS IT™ (M.I.Tech, Seoul, Republic of Korea) system is the most recent EC-LAMS introduced in the market. It is also a single-operator delivery EC-LAMS with inner diameters between 10 and 16 mm and a saddle length between flanges of 13 to 33 mm (Figure 1C). The delivery system is 10.5 Fr and the placement procedure is as follows: First, the sheath lock is unlocked. Then, the sheath handle is advanced with an electrocautery current to penetrate into the target lumen. After the stent is advanced under EUS visualization into the target lumen, the distal flange can be deployed by unlocking the device and rotating it 180°. As a next step, the distal deployed stent flange is pulled back to provide apposition of the luminal structures. For the deployment of the proximal flange of the stent, the device is rotated 180° again and pulled upwards.

## 3. Technical Considerations

Before performing EUS-GE, an imaging modality, preferably computed tomography (CT), should be performed to rule out the presence of bowel obstruction distal to the site of GOO and overt ascites, which would contraindicate the procedure. Since patients with GOO usually have significant amounts of gastric residue [3], pre-procedural management also includes “nil per mouth” 24 h before performing EUS-GE and placement of a large nasogastric tube to aspirate gastric content in order to minimize the risk of aspiration during the procedure [15]. Utilization of LAMSs for EUS-GE is off-label; thus, the patient should be advised and a proper consent specifically designed for EUS-GE needs to be explained and signed by the patient. Orotracheal intubation is recommended but not mandatory.

Nonetheless, even with the use of a LAMS, EUS-GE is considered by most experts a technically challenging procedure, given the fact that access to the target small bowel loop is difficult and unpredictable. Various techniques to perform EUS-GE based on adequately identifying and stabilizing the target jejunal loop have been described. These techniques can be divided into [16] device-assisted techniques, such as EUS-guided double-balloon-occluded gastrojejunostomy bypass (EPASS) [16,17]; direct techniques; and the wireless EUS-GE simplified technique (WEST) [18,19,20]. During EPASS, in which a dedicated balloon-occlusion catheter is advanced beyond the stricture, over a guidewire, into the proximal jejunum, both balloons are inflated and saline mixed with blue dye is instilled into the double-balloon-excluded enteral segment. In this way, adequate stabilization of the target loop is obtained, even though proper positioning of the dedicated balloon-occlusion device can be cumbersome and time-consuming. This excluded segment of jejunum is then targeted directly with the EC-LAMS without placement of a guidewire.

In the direct technique, the EUS linear scope is positioned to visualize the intended small bowel loop, which is then punctured with a 19G FNA needle and filled with saline mixed with contrast, before placing the EC-LAMS. When this technique was initially utilized, LAMS was inserted over a guidewire, which could result in displacement of the jejunum due to being pushed further away, leading to stent misdeployment. Thus, the use of a guidewire has mostly been abandoned. This technique can be used as a primary technique in case of a complete gastroduodenal obstruction not permitting passage of a guidewire.

In the WEST technique, a nasoenteric tube is advanced over a guidewire through the stricture into the small bowel distal to the GOO to fill the loop with saline and, using the nasobiliary catheter as a guide, LAMS placement is performed utilizing a free-hand technique (Figure 2). A variation of the WEST technique is possible when the stricture is short and incomplete; the small remaining stricture opening allows the injection of a 500–600 cc solution of saline, contrast, and blue dye by placing the tip of a standard endoscope into the stricture, with fluids instilled to distend the loop distal to the obstruction. The endoscope is then exchanged for an echoendoscope; once the proper loop is identified, it is punctured with a 22G needle to confirm the return of a blue solution and the needle is exchanged for a LAMS to complete the procedure (Figure 3).

There are only a few studies comparing procedural techniques. In a multicenter, retrospective study, technical, clinical, and adverse events were compared between the direct technique (52 patients) and the EPASS technique (22 patients). No differences or trends were found between the two groups in terms of important technical and clinical outcomes. The only difference was the procedural time, which was significantly longer for the EPASS versus the direct technique (89.9 ± 33.3 vs. 35.7 ± 32.1 min; *p* < 0.001) [21]. In another recent retrospective multicentric European study [22], the wireless endoscopic simplified technique (WEST) with an orointestinal drain was compared with the nonassisted direct technique over a guidewire (DTOG) in 71 patients (80% with GOO of malignant etiology). Technical success and clinical success rates at 1 month post-procedure were higher in the WEST group (95.1% vs. 73.3%, *p* = 0.01; 97.5% vs. 89.3%, *p* = ns.). The rate of AEs was also much lower in the WEST group (14.6% vs. 46.7%, *p* = 0.007). Thus, the authors concluded that the WEST technique should be preferred in the performance of EUS-GE, confirming previous observations that, by using a guidewire, the bowel loop could be pushed away with the catheter sheath of the device resulting in stent misdeployment [23].

Independently on the technique utilized to localize the proper loop for performing the anastomosis, LAMS placement is always achieved in the same way, with penetration of the wall of the stomach and the bowel loop utilizing a pure cut current at 100 W, effect 5. Caution should be made not to push the stent during this step to prevent the bowel loop from being driven further away, leading to stent misdeployment.

Once inside the bowel loop, the distal flange is opened under EUS view and then retracted back until it adheres to the bowel loop wall, at which time the proximal flange can be released under EUS, endoscopic, and/or fluoroscopic view based on the endoscopist’s preference. If the proximal flange is released inside the channel of the endoscope, finally the LAMS needs to be pushed out from the echoendoscope working channel, creating a gastroenteric anastomosis.

Before performing LAMS placement, administration of intravenous glucagon or buscopan to decrease bowel motility and small bowel emptying should be performed.

In case a blue dye has been instilled with saline and contrast, proper stent positioning can be proved by endoscopic visualization of blue-colored fluid into the stomach. At present, the lack of evidence regarding the superiority of any of the techniques utilized for EUS-GE prevents making any recommendation in favor of the use of one technique above the others [15].

Regarding which size of stent to use to create the anastomosis, a propensity score-matched study reported smaller-caliber LAMSs to be associated with a longer hospital stay and lower clinical success compared with large-caliber LAMSs. However, this finding was not confirmed in a multivariate analysis [19]. Similar results were found in a recent retrospective multicenter study on 267 patients in whom a 15 mm stent (148) or 20 mm stent (119) was placed [24]. Clinical success and AE rates were comparable, but a higher proportion of patients were able to tolerate a soft solid or complete diet (GOOS ≥ 2) in the 20 mm stent group vs. the 15 mm stent group (91.2% (84.4–95.7%) vs. 81.2% (73.9–87.2%), *p* = 0.04) [24]. In this regard, a recent meta-analysis including thirteen studies found that the 15 mm and 20 mm LAMS had similar pooled technical and clinical success rates (93.2% and 88.6% vs. 92.1% and 89.6%), with not statistically different AE rates of 11.4% vs. 14.7% [25]. However, an increased need for reintervention was noted for the 15 mm stents (pooled odds ratio, 3.59; 95% CI, 1.40–9.18, *p* = 0.008; reintervention rates of 10.3% vs. 3.5% for the 20 mm stents, respectively).

Another important issue is the learning curve needed to reach proficiency in EUS-GE. A multicenter retrospective study tried to answer this important question [26]. They evaluated EUS-GE results in 73 consecutive procedures performed by a single operator, utilizing the free-hand technique. Technical success was achieved in 95% of cases (68 of 73 patients), while peri-procedural AEs occurred in four patients (5.5%) with only one late AE (1%), all managed conservatively or endoscopically. All peri-procedural AEs occurred during the first 39 cases. Evaluation of the cumulative sum control chart curve revealed that performing at least 25 cases was needed in order to achieve proficiency and at least 40 cases for achieving mastery [26].

## 4. Adverse Events

The most frequent EUS-GE-related AEs are due to stent misdeployment (SM). This AE has been retrospectively evaluated in 467 patients from 18 centers, with all procedures performed using the direct or EPASS techniques. Stent misdeployment occurred in 46 patients (9.85%) and was classified into four types [27]. Type I, the most frequent one (63.1%), is characterized by distal flange deployment in the peritoneum and proximal flange release in the stomach, without evidence of a resulting enterotomy. This type of SM can be easily managed by removing the stent, followed by endoscopic closure of the gastrotomy site using through-the-scope or over-the-scope clips, or suturing devices. Only three out of the twenty-nine patients in whom this SM occurred underwent surgery because of signs of peritonitis (2) or to exclude a jejunal perforation (1). The second most frequent type of SM is type II (30.4%), in which the proximal flange is properly placed in the stomach, while the distal flange after visual confirmation on EUS of its penetration and release in the target jejunum migrates out into the peritoneum. This AE can be managed endoscopically with correct placement of another LAMS, or of a bridging, fully covered, tubular self-expanding metal stent (SEMS) through the initial misdeployed LAMS, but also with a simple removal of the stent with closure of gastrotomy that, in most cases, can avoid surgical intervention. Among the 14 patients who experienced this type of SM, only one underwent surgery because of early signs of peritonitis. The other two types of SM are far less frequent and characterized by correct placement of the distal flange into the jejunum with the release of the proximal flange into the peritoneum in Type III (2.2%), and creation of a gastro-colonic anastomosis in type IV (4.3%). Type III usually requires surgical intervention, while type IV can also be managed endoscopically with the closure of both sites, mainly with over-the-scope clips.

In the initial experience of EUS-GE, balloon dilatation of the LAMS after its placement was associated with a high risk of SM; thus, it is not performed anymore.

To increase the safety of EUS-GE, it would be important to standardize the procedure. Indeed, Park et al. [28] demonstrated that teaching EUS-GE using a standardized procedural protocol was associated with a significant decrease in AEs and an increase in technical success rate, irrespective of prior total experiences. In their study, the most experienced endoscopist in EUS-GE trained and supervised other advanced endoscopists with experience in EUS and EC-LAMS placement, but with no experience in EUS-GE. After different pre- and peri-procedural techniques were utilized (five procedures), a standardized protocol was introduced with the patient in a prone position, and the procedure was performed utilizing the WEST technique. Standardization of the procedure resulted in a significantly higher technical success rate (100% vs. 60%, *p* = 0.01) and significantly lower number of peri-procedural AEs (2.8% vs. 40%, *p* = 0.03). The direct technique, however, should only be reserved for those cases in which the bowel loop cannot be filled due to total closure of the stenotic tract, rendering this approach generally more difficult than others.

Regarding long-term complications, besides abdominal pain, gastrointestinal wall ulceration determined by the flange of the stent has also been reported [15]. So far, there are no concerns regarding perforation of the gastroenterostomy site during chemotherapy, including anti-VEGF agents, given the normal implication of growth factors in wound healing and previous reports of the association between anti-VEGF factors and gastrointestinal perforations. In this regard, the expansion force of the LAMS could in fact be useful in sealing off any perforation in the gastroenterostomy site; however, no clinical evidence exists to prove this.

## 5. EUS-Gastroenterostomy versus Surgical Gastro-Jejunostomy

Six studies, all with a retrospective design, compared the outcome of EUS-GE with that of surgical gastro-jejunostomy (SGJ) (Table 1) [19,29,30,31,32,33]. In the largest one, Canakis et al. [29] compared 187 EUS-GE with 123 SGJ procedures, of which 46 were performed laparoscopically. Overall, there was no difference in technical and clinical success rates between the two procedures, which were above 94% for both of them. In the EUS-GE arm, however, there were much fewer AEs, and the interval time to initiation/resumption of oral intake was much shorter than in the SGJ arm. In a subgroup analysis, EUS-GE was associated with a significantly shorter interval time to (re)initiation of chemotherapy (16.6 vs. 37.8 days, *p* < 0.001). The authors concluded that EUS-GE can be performed even among nutritionally deficient patients with the same efficacy as SGJ, resulting in fewer AEs and allowing earlier resumption of diet and oncological treatments. All studies reported so far in Table 1 display similar results to the study by Canakis et al. [29]. A recent systematic review comparing EUS-GE with SGJ (seven studies, 625 patients) showed lower technical success for EUS-GE, but higher per-protocol clinical success, and also lower overall AEs and shorter hospital stays compared with SGJ [34].

## 6. EUS-Gastroenterostomy versus Enteral Stenting

Three retrospective studies and a prospective registry study compared EUS-GE with enteral stenting for the treatment of malignant GOO (Table 1) [17,20,35,36]. In the largest retrospective study, van Wanrooij et al., [20] after propensity score matching, compared EUS-GE (88 patients) and enteral stenting (88 patients). No difference in technical success rates was found between the two arms, while clinical success rates were significantly higher in the EUS-GE arm (91% versus 75%, *p* = 0.008). Stent dysfunction occurred in 1% of patients in the EUS-GE arm versus 26% in the enteral stenting arm (*p* < 0.001). A trend towards an increased risk of AEs in the enteral stenting group was also observed (10% versus 21%, *p* = 0.09). These results suggest that EUS-GE might be superior and could be preferred over enteral stenting in patients with malignant GOO, where adequate expertise and resources are available.

Apart from the abovementioned study, in the only study with a prospective design, Vanella et al. presented the outcome of 70 patients undergoing EUS-GE performed via the wireless simplified technique [36]. Technical and clinical success rates were very high (97.1%), with a rate of AEs of 12.9%. Twenty-eight patients from this cohort were compared with twenty-eight in whom enteral stenting was performed. The comparison showed higher clinical success (100% vs. 75.0%, *p* = 0.006) which was achieved faster, and reduced recurrences (3.7% vs. 33.3%, *p* = 0.02) in the EUS-GE arm, along with a trend toward shorter time to the resumption of chemotherapy [36].

A recently accepted multicenter, randomized, controlled study by Teoh et al. [37] involved 97 patients with unresectable malignant GOO who underwent EUS-GE (48) using the EPASS technique or enteral stenting (49). There were no statistically significant differences in technical success and clinical success rates, 30-day mortality, 30-day AE, and QOL scores at 1 month. On the other hand, EUS-GE was associated with significantly lower 6-month reintervention rates (4.2% vs. 29%, *p* = 0.002, RR = 0.15 (95% CI: 0.04, 0.61)), longer mean stent patency (174.2 (70.9) vs. 147.9 (63.6) days, *p* = 0.013, HR = 0.13 (95% CI: 0.08, 0.22)), and a significantly better 1-month gastric outlet obstruction score (2.41 (0.7) vs. 1.91 (0.9), *p* = 0.012, r = 0.26).

## 7. Three-Way Comparisons (EUS-GE vs. Enteral Stenting vs. Surgical Gastroenterostomy)

Two retrospective studies compared the performance of all available techniques (EUS-GE vs. enteral stenting vs. surgical gastroenterostomy) for the palliation of malignant GOO [38,39]. In the larger study, Jaruvongvanich et al. [39] reported the experience in a cohort of 436 patients (232 EUS-GE, 131 enteral stenting, and 73 surgical GE). Technical success rates were above 98% in all study three arms. Conversely, clinical success rates were higher in the EUS-GE arm (98.3% vs. 91.6% and 90.4% for the enteral stenting and surgical GE arms, respectively; *p* < 0.0001). In addition, EUS-GE compared with both the enteral stenting and surgical-GE groups had a lower need for reintervention (0.9% vs. 12.2% and 13.7%, *p* < 0.0001), a shorter length of hospital stay (2 days vs. 3 days and 5 days, *p* < 0.0001), and lower AE rates (8.6% vs. 38.9% and 27.4%, *p* < 0.0001) [39].

In one of the most recent systematic reviews and meta-analyses, including 16 studies and 1541 patients [40], EUS-GE was associated with higher clinical success, without recurrent GOO, compared with enteral stenting (OR 5.08, 95%CI 3.42–7.55) but not SGJ. On the other hand, EUS-GE showed significantly lower rates of AEs compared with SGJ (OR 0.17, 95% CI 0.10–0.30), which were not significantly different versus enteral stenting alone (OR 0.57, 95% CI 0.29–1.14). These results were replicated by another comprehensive meta-analysis including twenty-six studies with 1493 patients, with the authors concluding that despite being technically challenging, EUS-GE has high technical and clinical success rates comparable with surgical gastroenterostomy, thus representing a very effective, minimally invasive procedure for GOO, once expertise in performing the procedure has been gained [41]. So far, no differences in outcomes of EUS-GE depending on the etiology of malignant GOO have been specifically reported.

## 8. Same-Session EUS-Guided Gastroenterostomy and Biliary Drainage

Obviously, an interventional endosonographer familiar with the techniques of EUS-guided gastroenterostomy and EUS-guided biliary drainage encounters the clinical scenario of a patient with both GOO and distal biliary stricture needing drainage, not amenable to standard endoscopic techniques. In the past, these patients were considered to have only a surgical option available, the so-called “double bypass”. However, in the last years, reports of successful minimally invasive treatments have been published [42]. These techniques were usually applied in separate endoscopic sessions. Until recently, same-session treatments have been reported in small series lacking a surgical comparison [43,44,45].

In a retrospective matched comparison involving five academic centers, 53 patients receiving same-session EUS-GE and biliary drainage by EUS-guided techniques were compared with 101 patients receiving double surgical bypass [46]. Technical success and clinical success rates (90.6% vs. 82.2%) were similar between the minimally invasive and surgical groups (96% and 91% vs. 100% and 82%, *p* > 0.05). Overall, AEs and severe adverse events (11.3% and 3.8% vs. 34.7% and 19.8%, *p* < 0.01) occurred more frequently in the surgical group, while in the EUS group, median times to oral intake and hospital stay (0 and 4 days vs. 6 and 13 days, respectively, *p* < 0.001) were significantly shorter. Noteworthily, similar to other studies, patients undergoing double drainage by EUS exhibited higher comorbidity scores, which made the authors conclude that despite being more severely ill, the patients undergoing same-session double EUS-guided achieved similar clinical success with fewer AEs compared with the surgical standard approach [46].

## 9. EUS-Gastroenterostomy for Benign Gastric Outlet Obstruction

Significant morbidity and nutritional compromise can occur in patients with benign GOO. Endoscopic treatment is associated with lower risk but is less durable compared with surgery. Unfortunately, surgery is associated with a higher morbidity, and this led to the exploration of other less-invasive methods to manage the condition. In a retrospective multicenter study evaluating EUS-GE in the management of 26 patients with benign GOO, the technical success rate was 96.2%, while after a median follow-up of 176.5 days (IQR: 47–445.8), the clinical success rate was 84%. Two adverse events due to stent misdeployment were noted [47]. Two subsequent studies demonstrated similar safety and efficacy [48,49]. More recently, Kahaleh et al. conducted a multicentre retrospective registry study comparison between benign and malignant indications for EUS-GE and noted no difference in outcomes [50]. EUS-GE, in the management of benign GOO, appears promising and may also have an additional role as a “bridging” procedure (e.g., for nutritional optimization) in patients who eventually require subsequent surgery.

## 10. Conclusions and Future Directions

Despite direct prospective comparative studies between EUS-GE versus endoscopic stenting and surgical gastro-jejunostomy still being limited, all available data described in this review strongly suggest EUS-GE as a very effective treatment of malignant GOO, with a good safety profile. Results of ongoing RCTs (ClinicalTrials.gov NCT05564143; NCT03259763; NCT05548114; NCT05561907; NCT05605327) will provide further evidence of the potential superiority of EUS-GE over the other available techniques, thus transforming this procedure as the standard of care for the treatment of malignant GOO.

Future studies with the main aim of standardizing the procedure, as well as specific training to increase the utilization of the procedure worldwide are necessary to render future data comparable and further increase the safety of the procedure. We strongly believe that EUS-guided gastroenterostomy is here to stay and will soon become the procedure of choice for the treatment of malignant and specific cases of benign GOO.

## Figures and Tables

**Figure 1 diagnostics-13-03308-f001:**
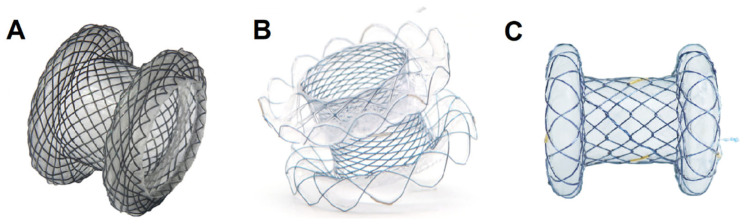
Lumen-apposing metal stents (LAMS) that can be used for EUS-guided gastroenterostomy: (**A**) the AXIOS™ stent (courtesy of Boston Scientific Corp., Marlborough, MA, USA); (**B**) the Spaxus™ stent (courtesy of Taewoong Medical Co., Ltd., Gimpo, Republic of Korea); (**C**) the Z-EUS IT™ stent (courtesy of M.I.Tech Co., Ltd., Pyongtaek, Republic of Korea) (reproduced from [12,13] with permission).

**Figure 2 diagnostics-13-03308-f002:**
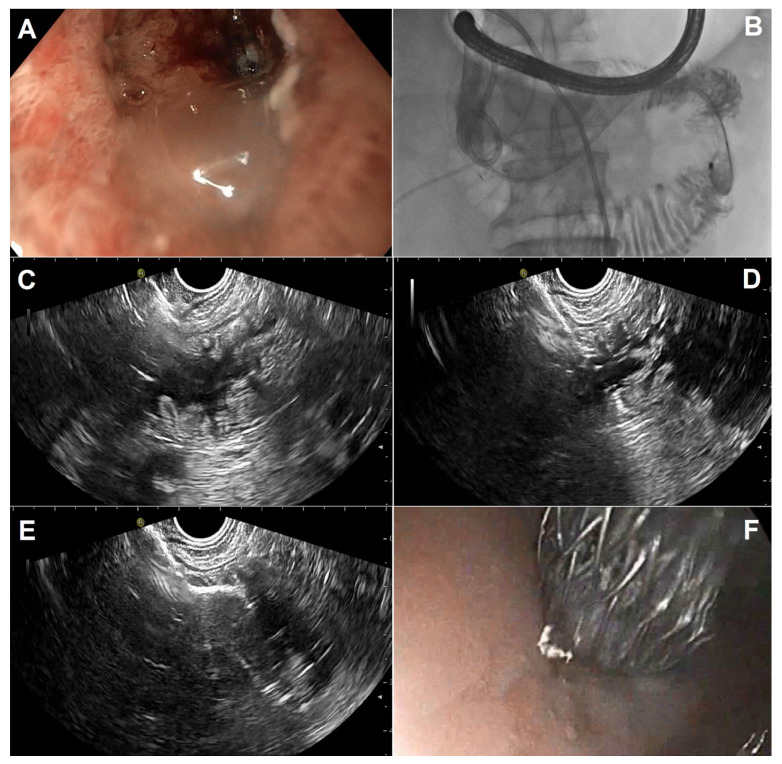
EUS-guided gastroenterostomy in a patient with duodenal stenosis from invasive pancreatic cancer previously treated by duodenal stenting using the WEST technique: (**A**) endoscopic image of a duodenal stent with tumoral ingrowth; (**B**) radiologic image depicting a 7 Fr nasobiliary catheter placed through the stricture after guidewire placement, through which saline and contrast are infused; (**C**) EUS identification of the jejunal target loop using fluid cavitation and identification of the nasobiliary catheter; (**D**) free-hand intrajejunal access with the electrocautery device (100 W, effect 5); (**E**) intrajejunal distal flange deployment under EUS guidance; (**F**) intragastric proximal flange deployment under endoscopic control.

**Figure 3 diagnostics-13-03308-f003:**
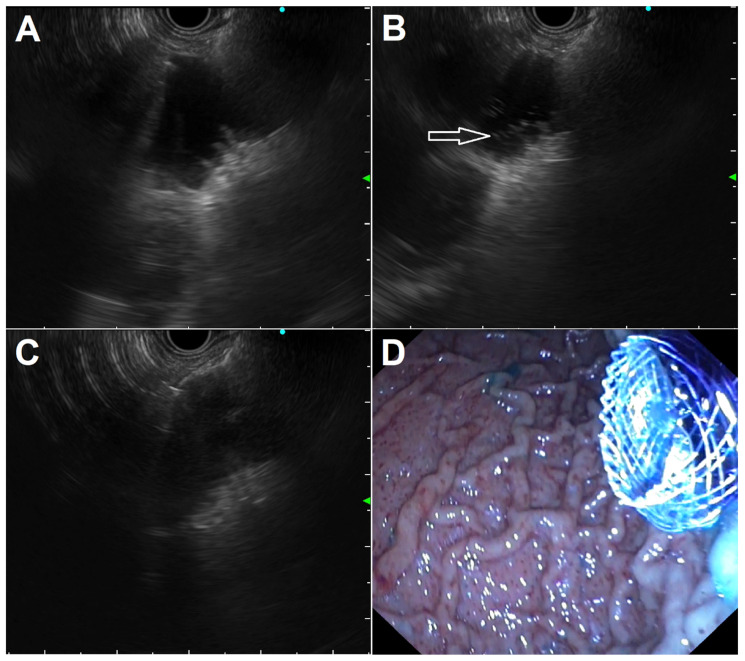
Standard steps in performing EUS-guided gastroenterostomy in a patient with a short and incomplete duodenal stricture leaving a small opening: (**A**) jejunal loop identification after its proper distension with saline mixed with contrast and methylene blue; (**B**) cystotome tip (arrow) advancement after cut with pure current for obtaining intrajejunal access; (**C**) distal flange deployment under EUS guidance; (**D**) endoscopic view of the proximal flange after deployment under EUS/endoscopic control, with spilling of blue-colored fluid indicating proper stent placement.

**Table 1 diagnostics-13-03308-t001:** Published studies comparing endoscopic ultrasound-guided gastroenterostomy (EUS-GE) with surgical gastroenterostomy or enteral stenting.

Author, Year	Design	EUS-GE *	Comparator	Technical Success	Clinical Success	Overall Adverse Events	Hospital Stay
**EUS-GE vs. surgery**							
Khashab et al., 2017 [30]	Retrospective	30 (100%)	63 (open)	87% vs. 100% (*p* = 0.009)	87% vs. 90% (*p* = 0.18)	16% vs. 25% (*p* = 0.3)	11.6 vs. 12.0 days (*p* = 0.35)
Perez-Miranda et al., 2017 [31]	Retrospective	25 (68%)	29 (lap)	88% vs. 100% (*p* = 0.11)	84% vs. 90% (*p* = 0.11)	12% vs. 41% (*p* = 0.038)	9.4 vs. 8.9 days (*p* = 0.75)
Bronswijk et al., 2021 [19]	Retrospective	77 (96%)	48 (lap)	95% vs. 100% (*p* = 0.297)	92% vs. 87.5% (*p* = 0.534)	6.5% vs. 31% (*p* < 0.001)	4.0 vs. 8.0 days (*p* < 0.001)
Kouanda et al., 2021 [32]	Retrospective	40 (76%)	26 (open)	92.5% vs. 100% (*p* = 0.15)	85% vs. 84% (*p* = 0.97)	N/a	5.0 vs. 14.5 days (*p* < 0.001)
Abbas et al., 2022 [33]	Retrospective	25 (100%)	27 (both open and lap)	100% in both groups	88% vs. 85% (*p* > 0.99)	8% vs. 41% (*p* = 0.01)	25.0 vs. 27.0 days
Canakis et al., 2023 [29]	Retrospective	187 (100%)	123 (46 lap)	98% vs. 100% (*p* = 0.15)	94% vs. 94% (*p* = 1.00)	13% vs. 33% (*p* < 0.001)	5.3 vs. 8.5 days (*p* < 0.001)
**EUS-GE vs. enteral stenting**							
Chen et al., 2017 [17]	Retrospective	30 (100%)	52	87% vs. 94% (*p* = 0.2)	83% vs. 67% (*p* = 0.12)	17% vs. 11.5% (*p* = 0.5)	11.3 vs. 9.5 days (*p* = 0.3)
Ge et al., 2019 [35]	Retrospective	22 (100%)	87	100% in both groups	96% vs. 76% (*p* = 0.042)	21% vs. 40% (*p* = 0.098)	7.4 vs. 7.9 days (*p* = 0.812)
Van Wanrooij et al., 2022 [20]	Retrospective	88 (100%)	88	94% vs. 98% (*p* = 0.44)	91% vs. 75% (*p* = 0.008)	10% vs. 20.5% (*p* = 0.09)	4.0 vs. 4.0 days (*p* = N/a)
Vanella et al., 2023 [36]	Prospective cohort study	28 (100%)	28	96% vs. 100% (*p* = 0.32)	100% vs. 75% (*p* = 0.006)	7% vs. 25% (*p* = 0.07)	6.5 vs. 7.0 days (*p* = 0.45)

* The percentage of malignant cause of gastric outlet obstruction is presented. N/a, not available; lap, laparoscopic.

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
