# Peer review of "The Role of Luminal Apposing Metal Stents on the Treatment of Malignant and Benign Gastric Outlet Obstruction"

_diagnostics, 2023, doi:10.3390/diagnostics13213308_

Round 1

Reviewer 1 Report

Please detail the eligibility and exclusion criteria for EUS-GE using LAMS for GOO. Is it also applicable for cases of ascites or peritoneal dissemination? Please state whether there is a difference in the clinical results of EUS-GE using LAMS for GOO caused by carcinoma (gastrointestinal cancer or pancreatobiliary tract cancer).

Minor editing of English language required

Author Response

Please detail the eligibility and exclusion criteria for EUS-GE using LAMS for GOO. Is it also applicable for cases of ascites or peritoneal dissemination? Please state whether there is a difference in the clinical results of EUS-GE using LAMS for GOO caused by carcinoma (gastrointestinal cancer or pancreatobiliary tract cancer).

RESPONSE

Indications and contraindications for EUS-GE have been added to the text of the paper. To the best of our knowledge, however, there is no study comparing outcomes of EUS-GE for different etiologies of GOO. This statement has been also added to the text of the paper.

Minor editing of English language required.

RESPONSE

The text has been once again reviewed by a native English speaker and minor corrections have been made.

Reviewer 2 Report

The authors review EUS-guided gastroenterostomy using luminal apposing metal stent for malignant and benign gastric outlet obstruction.

This manuscript is interesting and well written; however, it seems to be required some revisions.

1)    Although the title mentions malignant and benign gastric outlet obstruction, it would also be better to show the ratio of malignant and benign diseases in a table summarizing previous reports.

2)    In the introduction, only the demerit is emphasized for the enteral stenting and surgical gastrojejunostomy, and the merit should also be mentioned.

Minor

1)    Are there any concerns about perforation due to chemotherapy including bevacizumab after EUS-GE?

2)    There are some spacing errors (page 3, line 64, page 15, line 272 and page 17, line 314).

Author Response

The authors review EUS-guided gastroenterostomy using luminal apposing metal stent for malignant and benign gastric outlet obstruction.

This manuscript is interesting and well written; however, it seems to be required some revisions.

1)    Although the title mentions malignant and benign gastric outlet obstruction, it would also be better to show the ratio of malignant and benign diseases in a table summarizing previous reports.

RESPONSE

The percentage of malignant cases from the patients undergoing EUS-GE has been added to Table 1.

2)    In the introduction, only the demerit is emphasized for the enteral stenting and surgical gastrojejunostomy, and the merit should also be mentioned.

RESPONSE

The introduction section has been modified, as suggested.

Minor

1)    Are there any concerns about perforation due to chemotherapy including bevacizumab after EUS-GE?

RESPONSE

To the best of our knowledge, there has not been any report of perforation secondary to chemotherapy, including anti-VEGF medication, in cases of EUS-GE. A statement has been added to the text of the paper.

2)    There are some spacing errors (page 3, line 64, page 15, line 272 and page 17, line 314).

RESPONSE

These errors have been corrected.